# Coupled Variational Bayes via Optimization Embedding

*Bo Dai[1,2], *Hanjun Dai[1], Niao He[3], Weiyang Liu[1], Zhen Liu[1],
Jianshu Chen[4], Lin Xiao[5], Le Song[1,6]
[1]Georgia Institute of Technology, [2]Google Brain, [3]University of Illinois at Urbana Champaign
[4]Tencent AI, [5]Microsoft Research, [6]Ant Financial

## Abstract

Variational inference plays a vital role in learning graphical models, especially on large-scale datasets. Much of its success depends on a proper choice of auxiliary distribution class for posterior approximation. However, how to pursue an auxiliary distribution class that achieves both good approximation ability and computation efficiency remains a core challenge. In this paper, we proposed *coupled variational Bayes* which exploits the *primal-dual view* of the ELBO with the variational distribution class generated by an optimization procedure, which is termed *optimization embedding*. This flexible function class couples the variational distribution with the original parameters in the graphical models, allowing end-to-end learning of the graphical models by back-propagation through the variational distribution. Theoretically, we establish an interesting connection to gradient flow and demonstrate the extreme flexibility of this implicit distribution family in the limit sense. Empirically, we demonstrate the effectiveness of the proposed method on multiple graphical models with either *continuous* or *discrete* latent variables comparing to state-of-the-art methods.

## 1  Introduction

Probabilistic models with Bayesian inference provides a powerful tool for modeling data with complex structure and capturing the uncertainty. The latent variables increase the flexibility of the models, while making the inference intractable. Typically, one resorts to approximate inference such as sampling [Neal, 1993, Neal et al., 2011, Doucet et al., 2001], or variational inference [Wainwright and Jordan, 2003, Minka, 2001]. Sampling algorithms enjoys good asymptotic theoretical properties, but they are also known to suffer from slow convergence especially for complex models. As a result, variational inference algorithms become more and more attractive, especially driven by the recent development on stochastic approximation methods [Hoffman et al., 2013].

Variational inference methods approximate the intractable posterior distributions by a family of distributions. Choosing a proper variational distribution family is one of the core problems in variational inference. For example, the mean-field approximation exploits the distributions generated by the independence assumption. Such assumption will reduce the computation complexity, however, it often leads to the distribution family that is too restricted to recover the exact posterior [Turner and Sahani, 2011]. Mixture models and nonparametric family [Jaakkola and Jordon, 1999, Gershman et al., 2012, Dai et al., 2016a] are the natural generalization. By introducing more components in the parametrization, the distribution family become more and more flexible, and the approximation error is reduced. However, the computational cost increases since it requires the evaluations of the log-likelihood and/or its derivatives for each component in each update, which could limit the scalability of variational inference. Inspired by the flexibility of deep neural networks, many neural networks parametrized distributions [Kingma and Welling, 2013, Mnih and Gregor, 2014]

and tractable flows [Rezende and Mohamed, 2015, Kingma et al., 2016, Tomczak and Welling, 2016, Dinh et al., 2016] have been introduced as alternative families in variational inference framework. The compromise in designing neural networks for computation tractability restricts the expressive ability of the approximation distribution family. Finally, the introduction of the variational distribution also brings extra separate parameters to be learned from data. As we know, the more flexible the approximation model is, the more samples are required for fitting such a model. Therefore, besides the approximation error and computational tractability, the sample efficiency should also be taken into account when designing the variational distribution family.

In summary, most existing works suffer from a trade-off between approximation accuracy, computation efficiency, and sample complexity. It remains open to design a variational inference approach that enjoys all three aspects. This paper provides a method towards such a solution, called *coupled variational Bayes* (CVB). The proposed approach hinges upon two key components: **i)**, the *primal-dual* view of the ELBO; and **ii)**, the *optimization embedding* technique for generating variational distributions. The primal-dual view of ELBO avoids the computation of determinant of Jacobian in flow-based model and makes the arbitrary flow parametrization applicable, therefore, reducing the approximation error. The optimization embedding generates an interesting class of variational distribution family, derived from the updating rule of an optimization procedure. This distribution class reduces separate parameters by coupling the variational distribution with the original parameters in the graphical models. Therefore, we can back-propagate the gradient w.r.t. the original parameters through the variational distributions, which promotes the sample efficiency of the learning procedure. We formally justify that in continuous-time case, such a technique implicitly provides a flexible enough approximation distribution family from the gradient flow view, implying that the CVB algorithm also guarantees zero approximation error in the limit sense. These advantages are further demonstrated in our numerical experiments.

In the remainder of this paper, we first provide a preliminary introduction to problem settings described in directed graphical models and the variational auto-encoder (VAE) framework in Section 2. We present our coupled variational Bayes in Section 3, which leverages the optimization embedding in the primal-dual view of ELBO to couple the variational distribution with original graphical models. We build up the connections of the proposed method with the existing flows formulations in Section 4. We demonstrate the empirical performances of the proposed algorithm in Section 5.

## 2 Background

**Variational inference and learning**  Consider a probabilistic generative model, $p_\theta(x, z) = p_\theta(x|z)p(z)$, where $x \in \mathbb{R}^d$ denotes the observed variables and $z \in \mathbb{R}^r$ latent variables [2]. Given the dataset $\mathcal{D} = [x_i]_{i=1}^N$, one learns the parameter $\theta$ in the model by maximizing the marginal likelihood, *i.e.*, $\log \int p_\theta(x, z)dz$. However, the integral is intractable in general cases. Variational inference [Jordan et al., 1998] maximizes the evidence lower bound (ELBO) of the marginal likelihood by introducing an approximate posterior distribution, *i.e.*,

$$\log p_\theta(x) = \log \int p_\theta(x, z)dz \geqslant \mathbb{E}_{z \sim q_\phi(z|x)} \left[ \log p_\theta(x, z) - \log q_\phi(z|x) \right], \qquad (1)$$

where $\phi$ denotes the parameters of the variational distributions. There are two major issues in solving such optimization: **i)**, the appropriate parametrization for the introduced variational distributions, and **ii)**, the efficient algorithms for updating the parameters $\{\theta, \phi\}$. By adopting different variational distributions and exploiting different optimization algorithms, plenty of variants of variational inference and learning algorithms have been proposed. Among the existing algorithms, optimizing the objective with stochastic gradient descent [Hoffman et al., 2013, Titsias and Lázaro-gredilla, 2014, Dai et al., 2016a] becomes the dominated algorithm due to its scalability for large-scale datasets. However, how to select the variational distribution family has not been answered satifiedly yet.

**Reparametrized density**  Kingma and Welling [2013], Mnih and Gregor [2014] exploit the recognition model or inference network to parametrize the variational distributions. A typical inference network is a stochastic mapping from the observation $x$ to the latent variable $z$ with a set of global parameters $\phi$, *e.g.*, $q_\phi(z|x) := \mathcal{N}\left(z|\mu_{\phi_1}(x), \mathrm{diag}\left(\sigma^2_{\phi_2}(x)\right)\right)$, where $\mu_{\phi_1}(x)$ and $\sigma_{\phi_2}(x)$ are ofter

parametrized by deep neural networks. Practically, such reparameterizations have the closed-form of the entropy in general, and thus, the gradient computation and the optimization is relatively easy. However, such parameterization cannot perfectly fit the posterior when it does not fall into the known distirbution family, therefore, resulting extra approximation error to the true posterior.

**Tractable flows-based model**   Parameterizing the variational distributions with *flows* is proposed to mitigate the limitation of expressive ability of the variational distribution. Specifically, assuming a series of invertible transformations as $\{\mathcal{T}_t : \mathbb{R}^r \to \mathbb{R}^r\}_{t=1}^T$ and $z^0 \sim q_0(z|x)$, we have $z^T = \mathcal{T}_T \circ \mathcal{T}_{T-1} \circ \ldots \circ \mathcal{T}_1 (z^0)$ following the distribution $q_T(z|x) = q_0(z|x) \prod_{t=1}^T \left| \det \frac{\partial \mathcal{T}_t}{\partial z^t} \right|^{-1}$ by the change of variable formula. The flow-based parametrization generalizes the reparametrization tricks for the known distributions. However, a general parametrization of the transformation may violate the invertible requirement and result expensive or even infeasible calculation for the Jacobian and its determinant. Therefore, several carefully designed simple parametric forms of $\mathcal{T}$ have been proposed to compromise the invertible requirement and tractability of Jacobian [Rezende and Mohamed, 2015, Kingma et al., 2016, Tomczak and Welling, 2016, Dinh et al., 2016], at the expense of the flexibility of the corresponding variational distribution families.

# 3   Coupled Variational Bayes

In this section, we first consider the variational inference from a *primal-dual* view, by which we can avoid the computation of the determinant of the Jacobian. Then, we propose the *optimization embedding*, which generates the variational distribution by the adopt optimization algorithm. It automatically produces a *nonparametric* distribution class, which is flexible enough to approximate the posterior. More importantly, the optimization embedding couples the implicit variational distribution with the original graphical models, making the training more efficient. We introduce the key components below. Due to space limitation, we postpone the proof details of all the theorems in this section to Appendix A.

## 3.1   A Primal-Dual View of ELBO in Functional Space

As we introduced, the flow-based parametrization introduce more flexibility in representing the distributions. However, the calculating of the determinant of the Jacobian introduces extra computational cost and invertible requirement of the parametrization. In this section, we start from the primal-dual view perspective of ELBO, which will provide us a mechanism to avoid such computation and requirement, therefore, making the arbitrary flow parametrization applicable for inference.

As Zellner [1988], Dai et al. [2016a] show, when the family of variational distribution includes all valid distributions $\mathcal{P}$, the ELBO matches the marginal likelihood, *i.e.*,

$$L(\theta) := \mathbb{E}_{x \sim \mathcal{D}} \left[ \log \int p_\theta(x, z)\, dz \right] = \max_{q(z|x) \in \mathcal{P}} \underbrace{\mathbb{E}_{x \sim \mathcal{D}} \mathbb{E}_{z \sim q(z|x)} \left[ \log p_\theta(x|z) - KL\left(q(z|x)||p(z)\right) \right]}_{\ell_\theta(q)},$$

(2)

where $p_\theta(x, z) = p_\theta(x|z) p(z)$ and $\mathbb{E}_{x \sim \mathcal{D}}[\cdot]$ denotes the expectation over empirical distribution on observations and $\ell_\theta(q)$ stands for the objective for the variational distribution in density space $\mathcal{P}$ under the probabilistic model with $\theta$. Denote $q_\theta^*(z|x) := \mathrm{argmax}_{q(z|x) \in \mathcal{P}} \ell_\theta(q) = \frac{p_\theta(x,z)}{\int p_\theta(x,z) dz}$. The ultimate objective $L(\theta)$ will solely depend on $\theta$, *i.e.*,

$$L(\theta) = \mathbb{E}_{x \sim \mathcal{D}} \mathbb{E}_{z \sim q_\theta^*(z|x)} \left[ \log p_\theta(x, z) - \log q_\theta^*(z|x) \right],$$

(3)

which can be updated by stochastic gradient descent.

This would then require routinely solving the subproblem $\max_{q \in \mathcal{P}} \ell_\theta(q)$. Since the objective is taking over the whole distribution space, it is intractable in general. Traditionally, one may introduce special parametrization forms of distributions or flows for the sake of computational tractability, thus limiting the approximation ability. In what follows, we introduce an equivalent primal-dual view of the $\ell_\theta(q)$ in Theorem 1, which yields a promising opportunity to meet both approximation ability and computational tractability.

**Theorem 1 (Equivalent reformulation of $L(\theta)$)**   *We can reformulate the $L(\theta)$ equivalently as*

$$\min_{\nu \in \mathcal{H}_+} \mathbb{E}_{x \sim \mathcal{D}} \left[ \mathbb{E}_{\xi \sim p_\xi(\cdot)} \left[ \max_{z_{x,\xi} \in \mathbb{R}^r} \log p_\theta(x|z_{x,\xi}) - \log \nu(x, z_{x,\xi}) \right] + \mathbb{E}_{z \sim p(z)} \left[ \nu(x, z) \right] \right] - 1, \quad (4)$$

*where $\mathcal{H}_+ = \left\{ h : \mathbb{R}^d \times \mathbb{R}^r \to \mathbb{R}_+ \right\}$, $p_\xi(\cdot)$ denotes some simple distribution and the optimal $\nu_\theta^*(x, z) = \frac{q_\theta^*(z|x)}{p(z)}$.*

The primal-dual formulation of $L(\theta)$ is derived based on Fenchel-duality and interchangeability principle [Dai et al., 2016b, Shapiro et al., 2014]. With the primal-dual view of ELBO, we are able to represent the distributional operation on $q$ by local variables $z_{x,\xi}$, which provides an implicit *nonparametric* transformation from $(x,\xi) \in \mathbb{R}^d \times \Xi$ to $z_{x,\xi} \in \mathbb{R}^p$. Meanwhile, with the help of dual function $\nu(x,z)$, we can also avoid the computation of the determinant of Jacobian matrix of the transformation, which is in general infeasible for arbitrary transformation.

## 3.2 Optimization Embedding

In this section, inspired by the local variable representation of the variational distribution in Theorem 1, we will construct a special variational distribution family, which integrates the variational distribution $q$, *i.e.*, transformation on local variables, and the original parameters of graphical models $\theta$. We emphasize that optimization embedding is a general technique for representing the variational distributions and can also be accompanied with the original ELBO, which is provided in Appendix B.

As shown in Theorem 1, we switch handling the distribution $q(z|x) \in \mathcal{P}$ to each local variables. Specifically, given $x \sim \mathcal{D}$ and $\xi \sim p(\xi)$, with a fixed $\nu \in \mathcal{H}_+$,

$$z^*_{x,\xi;\theta} = \underset{z_{x,\xi} \in \mathbb{R}^p}{\text{argmax}} \ \log p_\theta(x|z_{x,\xi}) - \log \nu(x, z_{x,\xi}). \tag{5}$$

For the complex graphical models, it is difficult to obtain the global optimum of (5). We can approach the $z^*_{x,\xi}$ by applying mirror descent algorithm (MDA) [Beck and Teboulle, 2003, Nemirovski et al., 2009]. Specifically, denote the initialization as $z^0_{x,\xi}$, in $t$-th iteration, we update the variables until converges via the prox-mapping operator

$$z^t_{x,\xi;\theta} = \underset{z \in \mathbb{R}^r}{\text{argmax}} \ \left\langle z, \eta_t g\left(x, z^{t-1}_{x,\xi;\theta}\right)\right\rangle - D_\omega\left(z^{t-1}_{x,\xi;\theta}, z\right), \tag{6}$$

where $g\left(x, z^{t-1}_{x,\xi;\theta}\right) = \nabla_z \log p_\theta\left(x|z^{t-1}_{x,\xi;\theta}\right) - \nabla_z \log \nu\left(x, z^{t-1}_{x,\xi;\theta}\right)$ and $D_\omega(z_1, z_2) = \omega(z_2) - [\omega(z_1) + \langle \nabla\omega(z_1), z_2 - z_1\rangle]$ is the Bregman divergence generated by a continuous and strongly convex function $\omega(\cdot)$. In fact, we have the closed-form solution to the prox-mapping operator (6).

**Theorem 2 (The closed-form of MDA)** *Recall the $\omega(\cdot)$ is strongly convex, denote $f(\cdot) = \nabla\omega(\cdot)$, then, $f^{-1}(\cdot)$ exists. Therefore, the solution to (6) is*

$$z^t_{x,\xi;\theta} = f^{-1}\left(\eta_t g\left(x, z^{t-1}_{x,\xi;\theta}\right) + f\left(z^{t-1}_{x,\xi;\theta}\right)\right). \tag{7}$$

Proper choices of the Bregman divergences could exploit the geometry of the feasible domain and yield faster convergence. For example, if $z$ lies in the general continuous space, one may use $\omega(z) = \frac{1}{2}\|z\|_2^2$, the $D_\omega(\cdot, \cdot)$ will be Euclidean distance on $\mathbb{R}^r$, $f(z) = z$ and $f^{-1}(z) = z$, and if $z$ lies in a simplex, one may use $\omega(z) = \sum_{i=1}^r z_i \log z_i$, the $D_\omega(\cdot, \cdot)$ will be $KL$-divergence on the $p$-dim simplex, $f(z) = \log z$ and $f^{-1}(z) = \exp(z)$.

Assume we conduct the update (7) $T$ iterations, the mirror descent algorithm outputs $z^T_{x,\xi;\theta}$ for each pair of $(x,\xi)$. Therefore, it naturally establishes another *nonparametric* function that maps from $\mathbb{R}^d \times \Xi$ to $\mathbb{R}^r$ to approximate the sampler of the variational distribution point-wise, *i.e.*, $z^T_\theta(x,\xi) \approx z^*_{x,\xi;\theta}, \forall (x,\xi) \in \mathbb{R}^d \times \Xi$. Since such an approximation function is generated by the mirror descent algorithm, we name the corresponding function class as **optimization embedding**. Most importantly, the optimization embedded function explicitly depends on $\theta$, which makes the end-to-end learning possible by back-propagation through the variational distribution. The detailed advantage of using the optimization embedding for learning will be explained in Section 3.3.

Before that, we first justify the approximation ability of the optimization embedding by connecting to the gradient flow for minimizing the $KL$-divergence with a special $\nu(x,z)$ in the limit case. For simplicity, we mainly focus on the basic case when $f(z) = z$. For a fixed $x$, sample $\xi \sim p(\xi)$, the particle $z^T_\theta(x,\xi)$ is recursively constructed by transform $\mathcal{T}_x(z) = z + \eta g(x,z)$. We show that

**Theorem 3 (Optimization embedding as gradient flow)** *For a continuous time $t = \eta T$ and infinitesimal step size $\eta \to 0$, the density of the particles $z^t \in \mathbb{R}^r$, denoted as $q_t(z|x)$, follows nonlinear Fokker-Planck equation*

$$\frac{\partial q_t(z|x)}{\partial t} = -\nabla \cdot (q_t(z|x) g_t(x,z)), \tag{8}$$

*if $g_t(x,z) := \nabla_z \log p_\theta(x|z) - \nabla_z \log \nu^*_t(x,z)$ with $\nu^*_t(x,z) = \frac{q_t(z|x)}{p(z)}$. Such process defined by (8) is a gradient flow of $KL$-divergence in the space of measures with 2-Wasserstein metric.*

---

**Algorithm 1** Coupled Variational Bayes (CVB)

---

1: Initialize $\theta$, $V$ and $W$ (the parameters of $\nu$ and $z^0$) randomly, set length of steps $T$ and mirror function $f$.
   Set $z^0(x, \xi) = h_W(x, \xi)$.
2: **for** iteration $k = 1, \ldots, K$ **do**
3:     Sample mini-batch $\{x_i\}_{i=1}^m$ from dataset $\mathcal{D}$, $\{z_i\}_{i=1}^m$ from prior $p(z)$, and $\{\xi_i\}_{i=1}^m$ from $p(\xi)$.
4:     **for** iteration $t = 1, \ldots, T$ **do**
5:         Compute $z_\theta^t(x, \xi)$ for each pair of $\{x_i, \xi_i\}_{i=1}^m$.
6:         **Descend** $V$ with $\nabla_V \frac{1}{m} \sum_{i=1}^m \left[ \nu_V(x_i, z_i) - \log \nu_V(x_i, z_\theta^t(x_i, \xi_i)) \right]$.
7:     **end for**
8:     **Ascend** $\theta$ by stochastic gradient (11).
9:     **Ascend** $W$ by $\nabla_W \frac{1}{m} \sum_{i=1}^m \left[ \log p_\theta \left( x | z_\theta^T(x, \xi) \right) - \log \nu_V \left( x, z_\theta^T(x, \xi) \right) \right]$.
10: **end for**

---

From such a gradient flow view of optimization embedding, we can see that in limit case, the optimization embedding, $z_\theta^T(x, \xi)$, is flexible enough to approximate the posterior accurately.

### 3.3 Algorithm

Applying the optimization embedding into the $\ell_\theta(q)$, we arrive the approximate surrogate optimization to $L(\theta)$ in (2) as

$$\max_\theta \tilde{L}(\theta) := \min_{\nu \in \mathcal{H}_+} \mathbb{E}_{x \sim \mathcal{D}} \left[ \mathbb{E}_{\xi \sim p(\xi)} \left[ \log p_\theta \left( x | z_\theta^T(x, \xi) \right) - \log \nu \left( x, z_\theta^T(x, \xi) \right) \right] + \mathbb{E}_{z \sim p(z)} \left[ \nu(x, z) \right] \right].$$

$$(9)$$

We can apply the stochastic gradient algorithm for (9) with the unbiased gradient estimator as follows.

**Theorem 4 (Unbiased gradient estimator)** *Denote*

$$\nu_\theta^*(x, z) = \operatorname*{argmin}_{\nu \in \mathcal{H}_+} \mathbb{E}_{x \sim \mathcal{D}} \mathbb{E}_{z \sim p(z)} \left[ \nu(x, z) \right] - \mathbb{E}_{x \sim \mathcal{D}} \mathbb{E}_{\xi \sim p(\xi)} \left[ \log \nu \left( x, z_\theta^T(x, \xi) \right) \right], \qquad (10)$$

*we have the unbiased gradient estimator w.r.t. $\theta$ as*

$$
\begin{aligned}
\frac{\partial \tilde{L}(\theta)}{\partial \theta} &= \mathbb{E}_{x \sim \mathcal{D}} \mathbb{E}_{\xi \sim p(\xi)} \left[ \left. \frac{\partial \log p_\theta(x|z)}{\partial \theta} \right|_{z = z_\theta^T(x,\xi)} + \left. \frac{\partial \log p_\theta(x|z)}{\partial z} \right|_{z = z_\theta^T(x,\xi)} \frac{\partial z_\theta^T(x, \xi)}{\partial \theta} \right] \\
&\quad - \mathbb{E}_{x \sim \mathcal{D}} \mathbb{E}_{\xi \sim p(\xi)} \left[ \left. \frac{\partial \log \nu_\theta^*(x, z)}{\partial z} \right|_{z = z_\theta^T(x,\xi)} \frac{\partial z_\theta^T(x, \xi)}{\partial \theta} \right].
\end{aligned}
\qquad (11)
$$

As we can see from the gradient estimator (11), besides the effect on $\theta$ from the log-likelihood as in traditional VAE method with separate parameters of the variational distribution, which is the first term in (11), the estimator also considers the effect through the variational distribution explicitly in the second term. Such dependences through optimization embedding will potentially accelerate the learning in terms of sample complexity. The computation of the second term resembles to the back-propagation through time (BPTT) in learning the recurrent neural network, which can be easily implemented in Tensorflow or PyTorch.

**Practical extension**    With the functional primal-dual view of ELBO and the optimization embedding, we are ready to derive the practical CVB algorithm. The CVB algorithm can be easily incorporated with parametrization into each component to balance among approximation flexibility, computational cost, and sample efficiency. The introduced parameters can be also trained by SGD within the CVB framework. For example, in the optimization embedding, the algorithm requires the initialization $z_{x,\xi}^0$. Besides the random initialization, we can also introduce a parametrized function for $z_{x,\xi}^0 = h_W(x, \xi)$, with $W$ denoting the parameters. We can parametrize the $\nu(x, \xi)$ by deep neural networks with parameter $V$. To guarantee positive outputs of $\nu_V(x, \xi)$, we can use positive activation functions, *e.g.*, Gaussian, exponential, multi-quadratics, and so on, in the last layer. However, the neural networks parameterization may induce non-convexity, and thus, loss the guarantee of the global convergence in both (9) and (10), which leads to the bias in the estimator (11) and potential instability in training. Empirically, to reduce the effect from neural network parametrization, we update the parameters in $\nu$ within the optimization embedding simultaneously, implicitly pushing $z^T$ to follow the gradient flow. Taking into account of the introduced parameters, we have the CVB algorithm illustrated in Algorithm 1.

Moreover, we only discuss the optimization embedding through the basic mirror descent. In fact, other optimization algorithm, *e.g.*, the accelerated gradient descent, gradient descent with momentum, and other adaptive gradient method (Adagrad, RMSprop), can also be used for constructing the variational distributions. For the variants of CVB to parametrized continuous/discrete latent variables model and hybrid model with Langevin dynamics, please refer to the Appendix B and Appendix C.

## 4 Related Work

**Connections to Langevin dynamics and Stein variational gradient descent**   As we show in Theorem 3, the optimization embedding could be viewed as a discretization of a nonlinear Fokker-Plank equation, which can be interpreted as a gradient flow of $KL$-divergence on 2-Wasserstein metric with a special $\nu(x, z)$. It resembles the gradient flow with Langevin dynamics [Otto, 2001]. However, Langevin dynamics is governed by a linear Fokker-Plank equation and results a stochastic update rule, *i.e.*, $z^t = z^{t-1} + \eta \nabla \log p_\theta \left( x, z^{t-1} \right) + 2\sqrt{\eta} \xi^{t-1}$ with $\xi^{t-1} \sim \mathcal{N}(0, 1)$, thus different from our deterministic update given the initialization $z^0$.

Similar to the optimization embedding, Stein variational gradient descent (SVGD) also exploits a nonlinear Fokker-Plank equation. However, these two gradient flows follow from different PDEs and correspond to different metric spaces, thus also resulting different deterministic updates. Unlike optimization embedding, the SVGD follows interactive updates between samples and requires to keep a fixed number of samples in the whole process.

**Connection to adversarial variational Bayes (AVB)**   The AVB [Mescheder et al., 2017] can also exploit arbitrary flow and avoid the calculation related to the determinant of Jacobian via variational technique. Comparing to the primal-dual view of ELBO in CVB, AVB is derived based on classification density ratio estimation for $KL$-divergence in ELBO [Goodfellow et al., 2014, Sugiyama et al., 2012]. The most important difference is that CVB couples the adversarial component with original models through optimization embedding, which is flexible enough to approximate the true posterior and promote the learning sample efficiency.

**Connection to deep unfolding**   The optimization embedding is closely related to deep unfolding for inference and learning on graphical models. Existing schemes either unfold the *point estimation* through optimization [Domke, 2012, Hershey et al., 2014, Chen et al., 2015, Belanger et al., 2017, Chien and Lee, 2018], or expectation-maximization [Greff et al., 2017], or loopy BP [Stoyanov et al., 2011]. In contrast, the we exploit optimization embedding through a flow pointwisely, so that it handles the distribution in a *nonparametric* way and ensures enough flexibility for approximation.

## 5 Experiments

In this section, we justify the benefits of the proposed coupled variational Bayes in terms of the flexibility and the efficiency in sample complexity empirically. We also illustrate its generative ability. The algorithms are executed on the machine with Intel Core i7-4790K CPU and GTX 1080Ti GPUs. Additional experimental results, including the variants of CVB to discrete latent variable models and more results on real-world datasets, can be found in Appendix D. We The implementation is released at `https://github.com/Hanjun-Dai/cvb`.

### 5.1 Flexibility in Posterior Approximation

We first justify the flexibility of the optimization embedding in CVB on the simple synthetic dataset [Mescheder et al., 2017]. It contains $4$ data points, each representing a one-hot $2 \times 2$ binary image with non-zero entries at different positions. The generative model is a multivariate independent Bernoulli distribution with Gaussian distribution as prior, *i.e.*, $p_\theta(x|z) = \prod_{i=1}^4 \pi_i(z)^{x_i}$ and $p(z) = \mathcal{N}(0, I)$ with $z \in \mathbb{R}^2$, and $\pi_i(z)$ is parametrized by 4-layer fully-connected neural networks with $64$ hidden units in each latent layer. For

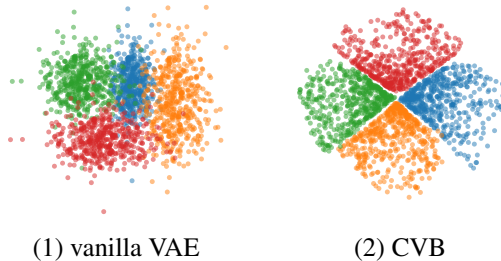

(1) vanilla VAE      (2) CVB

Figure 1: Distribution of the latent variables for VAE and CVB on synthetic dataset.

CVB, we set $f(z) = z$ in optimization embedding. We emphasize the optimization embedding is non-parametric and generated automatically via mirror descent. The dual function $\nu(x, z)$ is parametrized

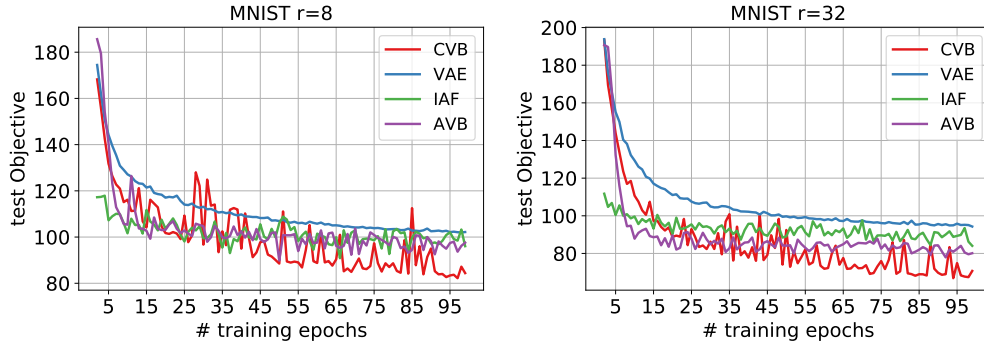

Figure 2: Convergence speed comparison in terms of number epoch on MNIST. We report the objective values of each method on held-out test set. The CVB achieves faster convergence speed comparing the other competitors in both $r = 8$ and $r = 32$ cases.

by a $(4 + 2)$-64-64-1 neural networks. The number of steps $T$ in optimization embedding is set to be 5 in this case.

To demonstrate the flexibility of the optimization embedding, we compare the proposed CVB with the vanilla VAE with a diagonal Gaussian posterior. A separate encoder in VAE is parametrized by reversing the structure of the decoder. We visualize the obtained posterior by VAE and CVB in Figure 1. While VAE generates a mixture of 4 Gaussians that is consistent with the parametrization assumption, the proposed CVB divides the latent space with a complex distribution. Clearly, this yields that CVB is more flexible in terms of approximation ability.

## 5.2   Efficiency in Sample Complexity

To verify the sample efficiency of CVB, we compare the performance of CVB on static binarize MNIST dataset to the current state-of-the-art algorithms, including VAE with inverse autoregressive flow (VAE+IAF) [Kingma et al., 2016], adversarial variational Bayes (AVB) [Mescheder et al., 2017], and the vanilla VAE with Gaussian assumption for the posterior distribution (VAE) [Kingma and Welling, 2013]. In this experiment, we use the Gaussian as the initialization in CVB. We follow the same setting as AVB [Mescheder et al., 2017], where conditional generative model $P(x|z)$ is a Bernoulli that is parameterized with 3-layer convolutional neural networks (CNN), and the inference model is also a CNN which is parametrized reversely as the generative model. Experiments for AVB and VAE+IAF are conducted based on the codes provided by Mescheder et al. [2017][3], where the default neural network structure are adopted. For all the methods, in each epoch, the batch size is set to be 100 while the initial learning rate is set to 0.0001.

We illustrate the convergence speed of testing objective values in terms of number epoch in Figure 2. As we can see, in both cases with the dimension of latent variable $r = 8$ and $r = 32$, the proposed CVB, represented by the red curve, converges to a lower test objective value in a much faster speed. We also compare the final approximated log likelihood evaluated by Importance Sampling, with the best baseline results reported in the original papers in Table 1. In this case, the objective function becomes too optimistic about the actual likelihood. It could

Table 1: The log-likelihood comparison between CVB and competitors on MNIST dataset. We can see that the proposed CVB achieves comparable performance on MNIST dataset.

| Methods | $\log p(x) \approx$ |
|---------|---------------------|
| CVB (8-dim) | -93.5 |
| CVB (32-dim) | -84.0 |
| AVB + AC (8-dim) | $-89.6$ [Mescheder et al., 2017] |
| AVB + AC (32-dim) | $-80.2$ [Mescheder et al., 2017] |
| DRAW + VGP | $-79.9$ [Tran et al., 2015] |
| VAE + IAF | $-79.1$ [Kingma et al., 2016] |
| VAE + NF ($T = 80$) | $-85.1$ [Rezende and Mohamed, 2015] |
| convVAE + HVI ($T = 16$) | $-81.9$ [Salimans et al., 2015] |
| VAE + HVI ($T = 16$) | $-85.5$ [Salimans et al., 2015] |

be caused by the Monte Carlo estimation of the Fenchel-Dual of $KL$-divergence, which is noisy comparing to the $KL$-divergence with closed-form in vanilla VAE. We can see that the proposed CVB still performs comparable with other alternatives. These results justify the benefits of parameters coupling through optimization embedding, especially in high dimension.

## 5.3 Generative Ability

We conduct experiments on real-world datasets, MNIST and CelebA, for demonstrating the generative ability of the model learned by CVB. For additional generated images, please refer to Appendix D.

**MNIST**  We use the model that is specified in Section 5.2. The generated images and reconstructed images by the variant of CVB in Appendix B.1 learned model versus the training samples are illustrated in the first row of Figure 3.

**CelebA**  We use the variant of CVB in Appendix B.4 to train a generative model with deep deconvolution network on CelebA-dataset for a 64-dimension latent space with $\mathcal{N}(0, 1)$ prior [Mescheder et al., 2017]. we use convolutional neural network architecture similar to DCGAN. We illustrate the results in the second row of Figure 3.

We can see that the learned models can produces realistic images and reconstruct reasonably in both MNIST and CelebA datasets.

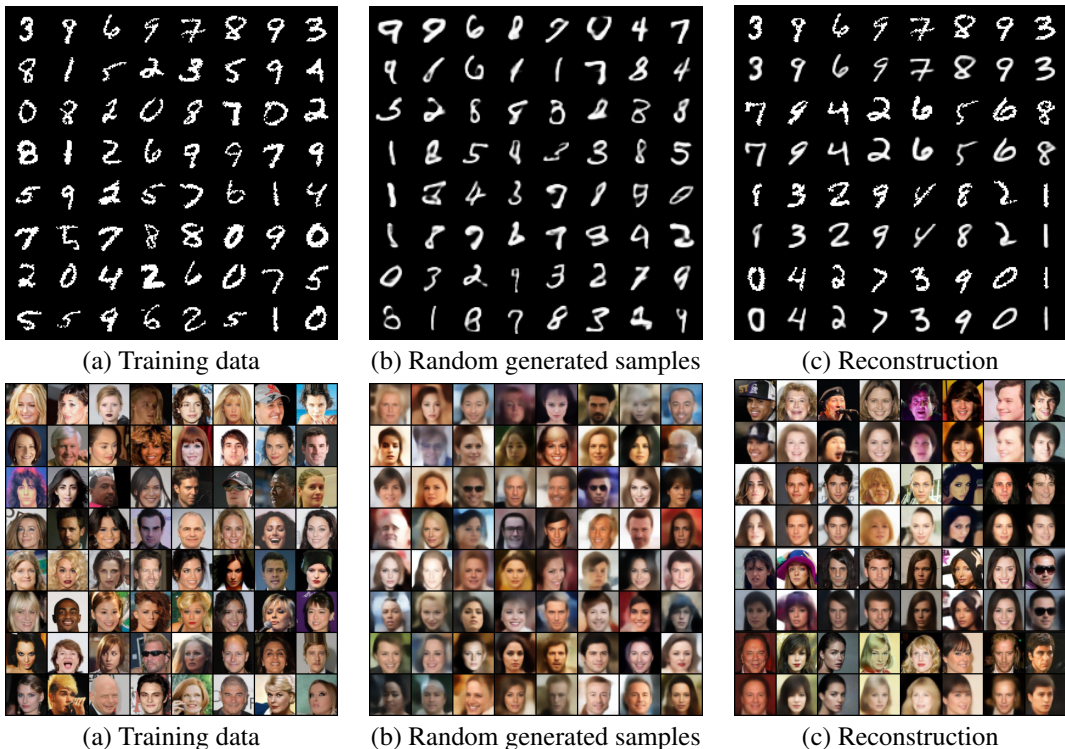

| (a) Training data | (b) Random generated samples | (c) Reconstruction |

| (a) Training data | (b) Random generated samples | (c) Reconstruction |

Figure 3: The training data, random generated images and the reconstructed images by the CVB learned models on MNIST and CelebA dataset. In the reconstruction column, the odd rows correspond to the test samples, and even rows correspond the reconstructed images.

## 6 Conclusion

We propose the coupled variational Bayes, which is designed based on the primal-dual view of ELBO and the optimization embedding technique. The primal-dual view of ELBO allows to bypass the difficulty with computing the Jacobian for non-invertible transformations and makes it possible to apply arbitrary transformation for variational inference. The optimization embedding technique, automatically generates a nonparametric variational distribution and couples it with the original parameters in generative models, which plays a key role in reducing the sample complexity. Numerical experiments demonstrates the superiority of CVB in approximate ability, computational efficiency, and sample complexity.

We believe the optimization embedding is an important and general technique, which is the first of the kind in literature and could be of independent interest. We provide several variants of the optimization embedding in Appendix B. It can also be applied to other models, *e.g.*, generative adversarial model and adversarial training, and deserves further investigation.

**Acknowledgments**

Part of this work was done when BD was with Georgia Tech. NH is supported in part by NSF CCF-1755829 and NSF CMMI-1761699. LS is supported in part by NSF IIS-1218749, NIH BIGDATA 1R01GM108341, NSF CAREER IIS-1350983, NSF IIS-1639792 EAGER, NSF IIS-1841351 EAGER, NSF CCF-1836822, NSF CNS-1704701, ONR N00014-15-1-2340, Intel ISTC, NVIDIA, Amazon AWS, Google Cloud and Siemens.

## Footnotes

*indicates equal contributions.

[2]We mainly discuss continuous latent variables in main text. However, the proposed algorithm can be extended to discrete latent variables easily as we show in Appendix B.

[3]The code can be found on `https://github.com/LMescheder/AdversarialVariationalBayes`.

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
