[Supplementary Material · supplementary.pdf]

# Appendix

## A    Proof of the Theorems in Section 3

We start by introducing the interchangeability principle Dai et al. [2016b], which plays a fundamental role for Theorem 1.

**Lemma 5 (interchangeability principle Dai et al. [2016b])** *Let $\xi$ be a random variable on $\Xi$ and assume for any $\xi \in \Xi$, function $g(\cdot, \xi) : \mathbb{R} \to (-\infty, +\infty)$ is a proper[4] and upper semicontinuous[5] concave function. Then*

$$\mathbb{E}_\xi[\max_{u \in \mathbb{R}} g(u, \xi)] = \max_{u(\cdot) \in \mathcal{G}(\Xi)} \mathbb{E}_\xi[g(u(\xi), \xi)].$$

*where $\mathcal{G}(\Xi) = \{u(\cdot) : \Xi \to \mathbb{R}\}$ is the entire space of functions defined on support $\Xi$.*

The result implies that one can replace the expected value of point-wise optima by the optimum value over a function space. More general results of interchange between maximization and integration can be found in [Rockafellar and Wets, 1998, Chapter 14] and [Shapiro et al., 2014, Chapter 7].

**Proof of Theorem 1:**    We apply the Fenchel dual form of $KL$-divergence, we have

$$KL(q\|p) = \left\langle q, \log \frac{q}{p} \right\rangle = \max_{\nu > 0} \langle q, \log \nu \rangle - \langle p, \nu \rangle + 1,$$

and

$$\nu^* = \underset{\nu > 0}{\operatorname{argmax}} \langle q, \log \nu \rangle - \langle p, \nu \rangle + 1 = \frac{q}{p}.$$

In fact, these equations are easy to verify by taking the gradient of the objective function and setting to zero. Plug such variational form into $\ell_\theta(q)$, we have

$$L(\theta) = \max_{q \in \mathcal{P}} \min_{\nu \in \mathcal{H}_+} \mathbb{E}_{x \sim \mathcal{D}} \left[ \mathbb{E}_q [\log p_\theta(x|z) - \log \nu(x, z)] + \mathbb{E}_{z \sim p(z)} [\nu(x, z)] \right] - 1, \qquad (12)$$

where $\mathcal{H}_+$ denotes the space which contains all positive functions, *i.e.*, $\mathcal{H}_+ = \{h : \mathbb{R}^d \times \mathbb{R}^r \to \mathbb{R}_+\}$. It is easy to verify (12) is concave-convex, therefore, the strong duality holds, which implies,

$$
\begin{aligned}
L(\theta) &= \min_{\nu \in \mathcal{H}_+} \max_{q \in \mathcal{P}} \mathbb{E}_{x \sim \mathcal{D}} \left[ \mathbb{E}_q [\log p_\theta(x|z) - \log \nu(x, z)] + \mathbb{E}_{z \sim p(z)} [\nu(x, z)] \right] - 1 \\
&= \min_{\nu \in \mathcal{H}_+} \max_{z(x,\xi) \in \mathcal{F}} \mathbb{E}_{x \sim \mathcal{D}} \left[ \mathbb{E}_{\xi \sim p(\xi)} [\log p_\theta(x|z(x, \xi)) - \log \nu(x, z(x, \xi))] + \mathbb{E}_{z \sim p(z)} [\nu(x, z)] \right] - 1,
\end{aligned}
$$

where the second equality comes from reparametrization and $\mathcal{F}$ denotes the transport mapping function space. In other words, as long as the $\mathcal{F}$ is flexible enough so that containing the function $z^*(x, \cdot)$ transform $p(\xi)$ to $q^*(z|x; \theta)$, the equality holds.

Under the mild condition that $\log p_\theta(x|\cdot)$ and $\log \nu(x, \cdot)$ are continuous, by applying the interchangeable principle in Lemma 5, we arrive the conclusion, *i.e.*,

$$L(\theta) = \min_{\nu \in \mathcal{H}_+} \mathbb{E}_{x \sim \mathcal{D}} \left[ \mathbb{E}_{\xi \sim p(\xi)} \left[ \max_{z_{x,\xi} \in \mathbb{R}^r} \log p_\theta(x|z_{x,\xi}) - \log \nu(x, z_{x,\xi}) \right] + \mathbb{E}_{z \sim p(z)} [\nu(x, z)] \right] - 1.$$

∎

**Proof of Theorem 2:**    We take the derivative to (6) w.r.t. $z$ and set to zero, resulting

$$
\begin{aligned}
& \eta_t g\left(x, z_{x,\xi;\theta}^{t-1}\right) - f(z) + f\left(z_{x,\xi;\theta}^{t-1}\right) = 0 \\
\Rightarrow \quad & z_{x,\xi;\theta}^t = f^{-1}\left(\eta_t g\left(x, z_{x,\xi;\theta}^{t-1}\right) + f\left(z_{x,\xi;\theta}^{t-1}\right)\right).
\end{aligned}
$$

The $f^{-1}$ exists due to the property of the strongly convexity of $\omega(\cdot)$.

∎

To prove Theorem 3, we first need the lemma Lemma 6, proved by Liu [2017], to guarantee the inconvertible of the transform.

**Lemma 6 ([Liu, 2017])** *Let $B$ be a square matrix and $\|B\|_F$ be the Forbenius norm. Let $\epsilon$ be a positive number such that $0 \leqslant \epsilon \leqslant \frac{1}{\rho(B+B^\top)}$, where $\rho(\cdot)$ denotes the spectrum radius. Then, $I + \epsilon\left(B + B^\top\right)$ is positive definite, and*

$$\log|\det(I + \epsilon B)| \geqslant \epsilon\operatorname{tr}(B) - \epsilon^2 \frac{\|B\|_F^2}{1 - \epsilon\rho(B + B^\top)}.$$

*Therefore, take an even small $\epsilon$ such that $0 \leqslant \epsilon \leqslant \frac{1}{2\rho(B+B^\top)}$, we get*

$$\log|\det(I + \epsilon B)| \geqslant \epsilon\operatorname{tr}(B) - 2\epsilon^2 \|B\|_F^2.$$

**Proof of Theorem 3:** The conclusion can be obtained by directly applying the Fokker-Planck Equation. We prove the result by infinitesimal analysis similar to Liu [2017][Appendix A.3].

For a fixed $x$, recall $\mathcal{T}_x(z) = z + \eta g(x, z)$, we denote $z \sim q(z|x)$. With a sufficient small $\eta$, $\nabla\mathcal{T}_x(z) = I + \eta\nabla g(x, z)$ is positive definite by lemma Lemma 6. Therefore, we have the inverse function of $\mathcal{T}_x^{-1}(z)$ as

$$\mathcal{T}_x^{-1}(z) = z - \eta g(x, z) + o(\eta).$$

The density of $\mathcal{T}_x(z)$ can be calculated by change of variables formula,

$$q'(z|x) = q\left(\mathcal{T}_x^{-1}(z)\,|x\right) \cdot \left|\det\left(\nabla\mathcal{T}_x^{-1}(z)\right)\right|.$$

Then, we have

$$
\begin{aligned}
\log q'(z|x) &= \log q\left(\mathcal{T}_x^{-1}(z)\,|x\right) + \log\left|\det\left(\nabla\mathcal{T}_x^{-1}(z)\right)\right| \\
&= \log q(z - \eta g(x, z)\,|x) + \log\det(z - \eta\nabla_z g(x, z) + o(\eta)) + o(\eta) \\
&= \log q(z|x) - \eta\nabla_z \log q(z|x)^\top g(x, z) - \eta\operatorname{tr}(\nabla_z g(x, z)) + o(\eta)
\end{aligned}
$$

where the third equation comes from Taylor expansion.

Therefore, by the definition of the derivative of $\log(\cdot)$, we have

$$
\begin{aligned}
\frac{q'(z|x) - q(z|x)}{\eta} &= \frac{q(z|x)(\log q'(z|x) - \log q(z|x))}{\eta} + o(\eta) \\
&= -q(z|x)\left(\eta\nabla_z \log q(z|x)^\top g(x, z) + \eta\operatorname{tr}(\nabla_z g(x, z))\right) + o(\eta) \\
&= -\nabla \cdot (q(z|x) g(x, z)) + o(\eta),
\end{aligned}
$$

which results the PDE as

$$\frac{\partial q_t(z|x)}{\partial t} = -\nabla \cdot (q_t(z|x) g_t(x, z)).$$

Recall $\nu_t(x, z) = \frac{q_t(z|x)}{p(z)}$ as proved in Theorem 1, we have the PDE as

$$
\begin{aligned}
\frac{\partial q_t(z|x)}{\partial t} &= -\nabla \cdot \left(q_t(z|x) \nabla\log\frac{p_\theta(x, z)}{q_t(z|x)}\right) \\
&= -\nabla \cdot (q_t(z|x) \nabla\log p_\theta(x, z)) + \Delta q_t(z|x) \\
&= -\frac{dKL(q_t(z|x)\,||p_\theta(x, z))}{dt} \\
&= \left\|\nabla\log\left(\frac{q_t(z|x)}{p_\theta(x, z)}\right)\right\|_{L_{q_t}}^2.
\end{aligned}
$$

Therefore, the PDE can be viewed as a gradient flow of $KL$-divergence under 2-Wasserstein metric [Otto, 2001].

∎

**Proof of Theorem 4:** The gradient estimator (11) can be directly obtained by applying the chain-rule with Danskin's theorem [Bertsekas, 1999]. We provide the derivation below for completeness.

$$
\begin{aligned}
\frac{\partial \tilde{L}(\theta)}{\partial \theta} &= \mathbb{E}_{x \sim \mathcal{D}} \mathbb{E}_{\xi \sim p(\xi)} \left[ \frac{\partial \log p_\theta(x|z)}{\partial \theta} \bigg|_{z=z_\theta^T(x,\xi)} + \frac{\partial \log p_\theta(x|z)}{\partial z} \bigg|_{z=z_\theta^T(x,\xi)} \frac{\partial z_\theta^T(x,\xi)}{\partial \theta} \right] \\
&- \mathbb{E}_{x \sim \mathcal{D}} \mathbb{E}_{\xi \sim p(\xi)} \left[ \frac{\partial \log \nu_\theta^*(x,z)}{\partial z} \bigg|_{z=z_\theta^T(x,\xi)} \frac{\partial z_\theta^T(x,\xi)}{\partial \theta} \right] \\
&+ \mathbb{E}_{x \sim \mathcal{D}} \mathbb{E}_{z \sim p(z)} \left[ \frac{\partial \nu_\theta^*(x,z)}{\partial \theta} \right] - \mathbb{E}_{x \sim \mathcal{D}} \mathbb{E}_{\xi \sim p(\xi)} \left[ \frac{\partial \log \nu_\theta^*(x,z)}{\partial \theta} \bigg|_{z=z_\theta^T(x,\xi)} \right].
\end{aligned}
$$

Denote $l(\theta) = \min_{\nu \in \mathcal{H}_+} \mathbb{E}_{x \sim \mathcal{D}} \mathbb{E}_{z \sim p(z)} [\nu(x,z)] - \mathbb{E}_{x \sim \mathcal{D}} \mathbb{E}_{\xi \sim p(\xi)} \left[ \log \nu \left( x, z_\theta^T(x,\xi) \right) \right]$, we can rewrite the third term as

$$
\frac{\partial l(\theta)}{\partial \nu} \bigg|_{\nu=\nu_\theta^*} \frac{\partial \nu}{\partial \theta} = \mathbb{E}_{x \sim \mathcal{D}} \mathbb{E}_{z \sim p(z)} \left[ \frac{\partial \nu_\theta^*(x,z)}{\partial \theta} \right] - \mathbb{E}_{x \sim \mathcal{D}} \mathbb{E}_{\xi \sim p(\xi)} \left[ \frac{\partial \log \nu_\theta^*(x,z)}{\partial \theta} \bigg|_{z=z_\theta^T(x,\xi)} \right].
$$

Recall the optimality of $\nu_\theta^*(x,z)$, it is easy to verify that $\frac{\partial l(\theta)}{\partial \nu} \bigg|_{\nu=\nu_\theta^*} = 0$, and thus, the third term in $\frac{\partial \tilde{L}(\theta)}{\partial \theta}$ is zero.

∎

# B  Several Variants of CVB

In Section 3, we mainly discussed the most general setting for $z_\theta^T(x,\xi)$, *i.e.*, we conduct optimization embedding for each pair of $(x,\xi) \in \mathbb{R}^d \times \Xi$ without any disributional form assumption. This provides the most flexible family for the variational distribution with the extra cost in fitting the dual function $\nu_V(x,z)$. In this section, we show several variants of CVB which are derived from applying optimization embedding to the posteriors for each pair with pre-fixed density forms, including Gaussian, categorical, and flow-based distributions in Section B.1, Section B.2, and Section B.3, respectively. In other words, we extend the optimization embedding under particular distribution assumption for *each* $q(z|x)$. We emphasize that it is still *nonparametric* since for each $x$, it owns a *separate* posterior. It is different from the vanilla amortized inference in VAE where the posterior parametrization are shared across all the samples. As we will see, these variants of CVB for pre-fixed parametric variational family will lead to Kim et al. [2018], Marino et al. [2018] as special cases of the framework. Finally, we apply the general optimization embedding technique to the parameters in the vanilla amortized VAE in Section B.4, resulting the parametric CVB.

These variants of CVB sacrifice the approximate ability for better computational efficiency, while still keep better sample complexity. In summary, we have the variants of CVB illustrated in Algorithm 2.

---
**Algorithm 2** CVB for Parametric Variational Posterior
---
1: Initialize $\theta$ and $W$ randomly, set length of steps $T$ and mirror update $f$.
2: **for** epoch $k = 1, \ldots, K$ **do**
3:     Sample mini-batch $\{x_i\}_{i=1}^m$ from dataset $\mathcal{D}$ and $\{\xi_i\}_{i=1}^m$ from $\mathcal{N}(0, \mathbf{I})$.
4:     Compute the $\left\{ z_\theta^T(x_i, \xi_i) \right\}_{i=1}^m$ via (16) for Gaussian latent variables, or via (22) for categorical latent variables, or via (27) for flow-based latent variables.
5:     Update $\theta$ and $W$ by stochastic gradient **ascend** with corresponding gradient estimator $\nabla_\theta \tilde{L}(\theta, W)$ and $\nabla_W \tilde{L}(\theta, W)$.
6: **end for**
---

## B.1  Optimization Embedding for Gaussian Latent Variables

We first illustrate applying the optimization embedding for continuous latent variables whose posterior is assumed as Gaussian, *i.e.*, $q(z|x) = \mathcal{N}\left(z|\phi_x, \text{diag}\left(\psi_x^2\right)\right)$ where $\{\phi_x, \psi_x\}$ denote the parameters depend on $x$. Therefore, we have $z(x,\xi) = \phi_x + \psi_x \cdot \xi$ with $\xi \sim \mathcal{N}(0, \mathbf{I})$. With such parametrization, the $KL$-divergence term in the ELBO will have closed-form, therefore, we do not need to introduce

the dual function $\nu(x, z)$. The EBLO, $L(\theta)$ becomes

$$\mathbb{E}_{x \sim \mathcal{D}} \mathbb{E}_{\xi \sim \mathcal{N}(0, \mathbf{I})} \left[ \underbrace{\max_{\phi_x, \psi_x} \log p_\theta(x | \phi_x + \psi_x \cdot \xi) + \frac{1}{2} \cdot \mathbf{1}^\top \left( 2 \log \psi_x - \phi_x^2 - \psi_x^2 \right)}_{\ell_\theta(\phi_x, \psi_x)} \right]. \quad (13)$$

Then, we can embed the optimization algorithm for $\phi_x, \psi_x$ to build up the connection between $z(x, \xi)$ with $\theta$. Specifically, we write out the updates for $\phi_x$ and $\psi_x$,

$$\phi_{x,\theta}^{t+1} = f^{-1} \left( \eta_t g_{\phi_x^t}(x, \theta) + f \left( \phi_{x,\theta}^t \right) \right), \quad (14)$$

$$\psi_{x,\theta}^{t+1} = f^{-1} \left( \eta_t g_{\psi_x^t}(x, \theta) + f \left( \psi_{x,\theta}^t \right) \right), \quad (15)$$

where $g_{\phi_x^t}(x, \theta)$ and $g_{\psi_x^t}(x, \theta)$ denote the gradient of $\ell_\theta(\phi_x, \psi_x)$ w.r.t. $\phi_x$ and $\psi_x$, respectively. We can also initialize the $\left[ \phi_{x,\theta}^0, \psi_{x,\theta}^0 \right] = h_W(x)$, where $W$ will be learned together by SGD. Therefore, after $T$ steps of the iteration, we have the function $z_\theta^T(x, \xi)$ from $\mathbb{R}^d \times \Xi$ to $\mathbb{R}^p$ as

$$z_\theta^T(x, \xi) = \phi_{x,\theta}^T + \psi_{x,\theta}^T \cdot \xi, \quad \xi \sim \mathcal{N}(0, \mathbf{I}). \quad (16)$$

Comparing the (16) with the most general (7), although $z_\theta^T(x, \xi)$ is still nonparametric in the sense $z$ changes individually for each pair $(x, \xi)$, the effect of the parametrization of posterior restricts the $z_\theta^T(x, \xi)$ to be a special form as derived in (16).

Plug (16) into $L(\theta)$, we have the surrogate objective $\tilde{L}(\theta, W)$ defined as

$$\mathbb{E}_{x \sim \mathcal{D}} \mathbb{E}_{\xi \sim \mathcal{N}(0, \mathbf{I})} \left[ \log p_\theta \left( x | z_\theta^T(x, \xi) \right) + \frac{1}{2} \cdot \mathbf{1}^\top \left( 2 \log \psi_{x,\theta}^T - \left( \phi_{x,\theta}^T \right)^2 - \left( \psi_{x,\theta}^T \right) \right)^2 \right].$$

Notice that in $\tilde{L}(\theta, W)$, we use optimization embedding cancel the max-operator on $\{\phi_x, \psi_x\}$. More importantly, we explicitly couple these parameters in the variational distributions with the parameter $\theta$ in the generative model. Then, we can apply the SGD for learning the $\theta$ and $W$. Similar to Theorem 4, we can derive the gradient estimator of $\theta$ as

$$\begin{aligned}
\frac{\partial \tilde{L}(\theta, W)}{\partial \theta} &= \mathbb{E}_{x \sim \mathcal{D}} \mathbb{E}_{\xi \sim \mathcal{N}(0, \mathbf{I})} \left[ \left. \frac{\partial \log p_\theta(x | z)}{\partial \theta} \right|_{z = z_\theta^T(x, \xi)} + \left. \frac{\partial \log p_\theta(x | z)}{\partial z} \right|_{z = z_\theta^T(x, \xi)} \frac{\partial z_\theta^T(x, \xi)}{\partial \theta} \right] \\
&+ \mathbb{E}_{x \sim \mathcal{D}} \mathbb{E}_{\xi \sim \mathcal{N}(0, \mathbf{I})} \left[ \frac{1}{2} \cdot \mathbf{1}^\top \left( \left( \frac{2}{\psi_{x,\theta}^T} - 2\psi_{x,\theta}^T \right) \frac{\partial \psi_{x,\theta}^T}{\partial \theta} - 2\phi_{x,\theta}^T \frac{\partial \phi_{x,\theta}^T}{\partial \theta} \right) \right] \quad (17)
\end{aligned}$$

As we can see, the first term in (17) and (11) is the same, while the second term is different. Due to the closed-form of $KL$-divergence in Gaussian parametrization, we can calculate the gradient w.r.t. $\theta$ of the $KL$-divergence term in ELBO directly as in (17), while in the general case (11), we need to fit the $\nu(x, \xi)$ for approximating the $KL$-divergence.

## B.2 Optimization Embedding for Categorical Latent Variables

Similarly, the optimization embedding can also be applied to categorical latent variables models. To ensure the gradient is valid, we approximate the categorical latent variables with Gumbel-Softmax [Jang et al., 2016, Maddison et al., 2016], *i.e.*,

$$q_\phi(z|x) = \Gamma(r) \tau^{r-1} \left( \sum_{i=1}^r \frac{\pi_{x,\phi,i}}{z_i^\tau} \right)^{-r} \prod_{i=1}^r \left( \frac{\pi_{x,\phi,i}}{z_i^{\tau+1}} \right), \quad (18)$$

and

$$z_{\phi,i}(x, \xi) = \frac{\exp\left( (\phi_{x,i} + \xi_i) / \tau \right)}{\sum_{i=1}^r \exp\left( (\phi_{x,i} + \xi_i) / \tau \right)}, \quad \xi_i \sim \mathcal{G}(0, 1), \quad i \in \{1, \ldots, r\}, \quad (19)$$

with $\pi_{x,\phi,i} = \frac{\exp(\phi_{x,i})}{\sum_{i=1}^{p}\exp(\phi_{x,i})}$ and $\mathcal{G}(0,1)$ denotes the Gumbel distribution. Denote the initialization function parametrized by $W$, follow the same derivation, we have the ELBO, $L(\theta)$, as

$$\mathbb{E}_{x \sim \mathcal{D}}\mathbb{E}_{\xi \sim \mathcal{G}(0,\mathbf{I})}\left[\max_{\phi_x}\ \underbrace{\log p_\theta\left(x, z_\phi\left(x,\xi\right)\right) + p\log\left(\mathbf{1}^\top\frac{\pi_{x,\phi}}{\left(z_\phi\left(x,\xi\right)\right)^\tau}\right) - \mathbf{1}^\top\log\frac{\pi_{x,\phi}}{\left(z_\phi\left(x,\xi\right)\right)^{\tau+1}}}_{\ell_\theta(\phi_x)}\right].$$

(20)

Similarly, we embed the optimization procedure for $\phi_x$, resulting

$$\phi_{x,\theta}^{t+1} = f^{-1}\left(\eta_t g_{\phi_{x,\theta}^t}\left(x,\theta\right) + f\left(\phi_{x,\theta}^t\right)\right),$$

(21)

$g_{\phi_{x,\theta}^t}\left(x,\theta\right)$ denotes the gradient of $\ell_\theta\left(\phi_x\right)$ w.r.t. $\phi$. Therefore, after $T$ steps of the iteration, we have the function $z_\theta^T\left(x,\xi\right)$ from $\mathbb{R}^d \times \Xi$ to $\mathbb{R}^r$ as

$$z_{\theta,i}^T\left(x,\xi\right) = \frac{\exp\left(\left(\phi_{x,\theta,i}^T + \xi_i\right)/\tau\right)}{\sum_{i=1}^{p}\exp\left(\left(\phi_{x,\theta,i}^T + \xi_i\right)/\tau\right)}, \quad \xi_i \sim \mathcal{G}\left(0,1\right), \quad i \in \{1, \ldots, r\},$$

(22)

where $\phi_{x,\theta}^0 = h_W\left(x\right)$. Plug (22) into $L\left(\theta\right)$, we have the surrogate objective $\tilde{L}\left(\theta\right)$ defined as

$$\mathbb{E}_{x \sim \mathcal{D}}\mathbb{E}_{\xi \sim \mathcal{G}(0,\mathbf{I})}\left[\log p_\theta\left(x, z_\theta^T\left(x,\xi\right)\right) + p\log\left(\mathbf{1}^\top\frac{\pi_{\phi_{x,\theta}^T}}{\left(z_\theta^T\left(x,\xi\right)\right)^\tau}\right) - \mathbf{1}^\top\log\frac{\pi_{\phi_{x,\theta}^T}}{\left(z_\theta^T\left(x,\xi\right)\right)^{\tau+1}}\right].$$

We can follow the chain-rule to calculate the gradient estimator w.r.t. $\theta$ and $W$ for $\tilde{L}\left(\theta,W\right)$, and apply the SGD to optimize $\tilde{L}\left(\theta,W\right)$ w.r.t. both $\theta$ and $W$.

## B.3 Optimization Embedding for Flow-based Latent Variables

In this section, we derive the optimization embedding for the distributions generated by change-of-variables, *i.e.*, flow-based models. Specifically, we assume that the latent variables follow the distribution generated by change of variables, *i.e.*,

$$q_\phi\left(z|x\right) = p\left(\mu_{\phi_x}^{-1}\left(z\right)\right)\left|\det\frac{\partial\mu_{\phi_x}^{-1}\left(z\right)}{\partial z}\right|.$$

(23)

where $\mu_{\phi_x}\left(\cdot\right)$ denotes the bijective function and $p\left(\cdot\right)$ denotes some simple distribution. Then, we can sample $z$ as

$$z_\phi\left(x,\xi\right) = \mu_{\phi_x}\left(\xi\right), \quad \xi \sim p\left(\xi\right).$$

(24)

It is easy to see that the Gaussian and categorical distributed latent variable are just special cases of the general change-of-variables. There are several carefully designed simple parametric forms of $\mu_{\phi_x}$ have been proposed to compromise the invertible requirement and tractability of Jacobian, *e.g.*, normalizing flow [Rezende and Mohamed, 2015, Tomczak and Welling, 2016], autoregressive flow [Kingma et al., 2016], and partition flow [Dinh et al., 2016]. The optimization embedding can be applied to all of these parametrizations. Follow the same derivation, we have the ELBO, $L\left(\theta\right)$, as

$$\mathbb{E}_{x \sim \mathcal{D}}\mathbb{E}_{\xi \sim p(\xi)}\left[\max_{\phi_x}\underbrace{\log p_\theta\left(x, \mu_{\phi_x}\left(\xi\right)\right) + \log p\left(\xi\right) - \log\left|\det\frac{\partial\mu_{\phi_x}\left(\xi\right)}{\partial\xi}\right|}_{\ell_\theta(\phi_x)}\right].$$

(25)

Similarly, we embed the optimization procedure for the $\phi_x$, resulting

$$\phi_{x,\theta}^{t+1} = f^{-1}\left(\eta_t g_{\phi_{x,\theta}^t}\left(x,\theta\right) + f\left(\phi_{x,\theta}^t\right)\right),$$

(26)

$g_{\phi_{x,\theta}^t}\left(x,\theta\right)$ denotes the gradient of $\ell_\theta\left(\phi_x\right)$ w.r.t. $\phi$. Therefore, after $T$ steps of the iteration, we have the function $z_\theta^T\left(x,\xi\right)$ from $\mathbb{R}^d \times \Xi$ to $\mathbb{R}^r$ as

$$z_\theta^T\left(x,\xi\right) = \mu_{\phi_x^T}\left(\xi\right), \quad \xi \sim p\left(\xi\right),$$

(27)

where $\phi_x^0 = h_W(x)$. Plug (27) into $L(\theta)$, we have the surrogate objective $\tilde{L}(\theta, W)$ defined as

$$\mathbb{E}_{x \sim \mathcal{D}} \mathbb{E}_{\xi \sim p(\xi)} \left[ \log p_\theta \left( x, \mu_{\phi_x^T}(\xi) \right) + \log p(\xi) - \log \left| \det \frac{\partial \mu_{\phi_x}(\xi)}{\partial \xi} \right| \Bigg|_{\phi_x = \phi_x^T} \right],$$

where $\log \left| \det \frac{\partial \mu_{\phi_x}(\xi)}{\partial \xi} \right| \Big|_{\phi_x = \phi_x^T}$ denotes the log-determinant value of $\frac{\partial \mu_{\phi_x}(\xi)}{\partial \xi}$ with $\phi_x$ set to be $\phi_x^T$.

We can follow the chain-rule to calculate the gradient estimator w.r.t. $\theta$ and $W$ for $\tilde{L}(\theta, W)$, and apply the SGD to optimize $\tilde{L}(\theta, W)$ w.r.t. both $\theta$ and $W$.

### B.4 Parametric Optimization Embedding

We have demonstrated the optimization embedding to generate the nonparametric variational distributions, either in the form of arbitrary flow or individual pre-fixed distribution for each sample. In fact, the optimization embedding is so general that its stochastic variant can be even applied to the parameters of the vanilla amortized VAE. Specifically, we take $q_\phi(z|x) = \mathcal{N}\left( \mu_{\phi_1}(x), \text{diag}\left( \sigma_{\phi_2}^2(x) \right) \right)$ as an example, the ELBO becomes

$$\max_\phi \mathbb{E}_{x \sim \mathcal{D}} \mathbb{E}_{\xi \sim \mathcal{N}(0, \mathbf{I})} \left[ \log p_\theta \left( x | \mu_{\phi_1}(x) + \sigma_{\phi_2}(x) \cdot \xi \right) + \frac{1}{2} \cdot \mathbf{1}^\top \left( 2 \log \sigma_{\phi_2}(x) - \mu_{\phi_1}(x) - \sigma_{\phi_2}^2(x) \right) \right].$$
(28)

Since the variable $\phi$ is global, the calculation of the gradient w.r.t. $\phi$ requires visiting the whole dataset. However, we can use stochastic gradient. Then, the stochastic optimization procedure for global parameter $\phi$ from $\phi^0$ can be embedded as

$$\phi_\theta^{t+1} = f^{-1}\left( \eta_t \widehat{g}_{\phi_\theta^t}(\theta) + f\left( \phi_\theta^t \right) \right),$$
(29)

where $\widehat{g}$ denote the stochastic approximation of the true gradient. Plug $\phi^T(\theta)$ into $L(\theta, \phi^0)$, we have the surrogate objective $\tilde{L}(\theta, \phi^0)$ defined as

$$\mathbb{E}_{x \sim \mathcal{D}} \mathbb{E}_{\xi \sim \mathcal{N}(0, \mathbf{I})} \left[ \log p_\theta \left( x | \mu_{\phi_{1,\theta}^T}(x) + \sigma_{\phi_{2,\theta}^T}(x) \cdot \xi \right) + \frac{1}{2} \cdot \mathbf{1}^\top \left( 2 \log \sigma_{\phi_{2,\theta}^T}(x) - \mu_{\phi_{1,\theta}^T}(x) - \sigma_{\phi_{2,\theta}^T}^2(x) \right) \right].$$

We can follow the chain-rule to calculate the gradient estimator for $\tilde{L}(\theta, \phi^0)$, and apply the SGD to optimize w.r.t. $\theta$ and $\phi^0$.

The significant difference between the parametric CVB and the CVB with Gaussian latent variable in Section B.1 is the optimization embedding objects: the former is w.r.t. the *global* amortized variational distribution parameters, while the latter one is w.r.t. the *local* variables for each sample $x$. Meanwhile, the key difference between parametric CVB and the vanilla amortized VAE is that the calculation of the gradient w.r.t. $\theta$ now need to back-propagate through $\phi_\theta^T$. With the increasing of $T$, the computation cost will be increasing. Therefore, in practice, one needs to balance the embedding accuracy and the computational cost by tuning $T$.

## C  Hybrid CVB

As we discussed in Section 4, the optimization embedding with optimal dual and the Langevin dynamics are all follows the gradient flow in 2-Wasserstein metric, just with different Fokker-Plank equations. Therefore, it is nature to combine the proposed optimization embedding with Langevin dynamcis, therefore, we have a stochastic mapping from $\mathbb{R}^d \times \Xi \to \mathbb{R}^r$ as

$$z_{x, \xi^0; \theta}^t = (1 - \lambda) f^{-1}\left( \eta_t g\left( x, z_{x, \xi^0; \theta}^{t-1} \right) + f\left( z_{x, \xi^0; \theta}^{t-1} \right) \right) \tag{30}$$

$$+ \lambda \left( z_{x, \xi; \theta}^{t-1} + \eta_t \nabla \log p_\theta(x, z) + 2\sqrt{\eta} \xi^{t-1} \right), \tag{31}$$

where $\xi^i \sim \mathcal{N}(0, I)$ and $\lambda \in \{0, 1\}$.

We can replace the optimization embedding in Algorithm 1 with the hybrid embedding, which achieves the hybrid CVB.

Figure 4: Convergence speed comparison in terms of number epoch on MNIST for discrete latent variable models. The CVB achieves better test ELBO with faster convergence speed comparing to the original Gumbel-Softmax parametrization for both categorial and Bernoulli distributed latent variables.

# D  Additional Experiments

## D.1  Generality on Discrete Latent Variable Models

We test the CVB on categorical and binary latent variable model on MNIST. We utilize the Gumbel-Softmax to relax the distribution over the discrete latent variables and apply the optimization embedding variant introduced in Section B.2. We conduct comparison between the CVB on discrete latent variable models with the VAE with Gumbel-Softmax reparametrization trick [Jang et al., 2016], which is the current state-of-the-art. The results are illustrated in Figure 4. We can see that the parametrized CVB achieves better performance in a faster speed.

## D.2  Additional Results on Generative Ability

The additional experimental results on MNIST and CelebA via the variants of CVB in Appendix B.1 and Appendix B.4, respectively, are illustrated in Figure 5 and Figure 6.

Figure 5: Generated images for MNIST dataset by CVB.

Figure 6: Generated images for CelebA dataset by CVB.

## Footnotes

[4]We say $g(\cdot, \xi)$ is proper when $\{u \in \mathbb{R} : g(u, \xi) < \infty\}$ is non-empty and $g(u, \xi) > -\infty$ for $\forall u$.

[5]We say $g(\cdot, \xi)$ is upper semicontinuous when $\{u \in \mathbb{R} : g(u, \xi) < \alpha\}$ is an open set for $\forall \alpha \in \mathbb{R}$. Similarly, we say $g(\cdot, \xi)$ is lower semicontinuous when $\{u \in \mathbb{R} : g(u, \xi) > \alpha\}$ is an open set for $\forall \alpha \in \mathbb{R}$.