[Reviews · NeurIPS 2018]

Reviewer 1



The authors derive a dual representation of the ELBO and leverage it do develop an original and interesting "particle flow" algorithm for fitting latent variable generative models. Comments: ======== 1) Proof of Theorem is quite straightforward and the main novelty of the paper -- I think that the proof should be clarified and included in the main text. At the moment, the proof is not up to the standard of a NIPS paper. For example: * there is no KL term in the original definition of \ell_\theta while the proof of Theorem 1 starts by applying Fenchel duality to KL * the authors then use the bracket notations for expectations and then switch back the \mathbb{E}[...] notation. * I do not think that the notation \mathcal{F} has been defined. All this can be understood with some effort, but I think this should be clarified 2) The sentence "More importantly, the optimization embedding couples the implicit variational distribution with the original graphical models, making the training more efficient." is very important, and I think that it deserves more comments / clarifications. 3) For clarity of exposition, is it really necessary to talk about Mirror Descent while it seems that most of the experiments are done using vanilla gradient descent. 4) The text is very difficult to read -- notations are not always defined and/or optimal, and not much effort has been put into polishing the text. This is a pity since the idea is interesting! There are many unnecessary other parts of text that could be deleted without hurting the clarity & flow of the paper.

Reviewer 2



The paper proposes a new distribution class that can be used as an auxiliary distribution class for variational inference. This new class is called 'Optimization embedding' and is derived based on two steps: first a primal-dual rewriting of ELBO, and second, and optimization procedure based on the minor descent algorithm. a Particular instance of this distribution class is described that can be trained using gradient-decent techniques. The model is evaluated among three axis: the generative ability of the approach, the flexibility of the resulting approximation posterior distribution and sample efficiency. These experiments are made on classical datasets (including toy datasets, MNIST and CelebA) and show that the proposed class has good properties. First of all, the model presented in the paper is interesting and is a good contribution to the field. The paper is very technical and very difficult to understand for a non-expert. It needs a strong background on different methods (e.g normalizing flows, Fenchel-duality and interchangeability principle, mirror descent algorithm, etc...) An effort could be put on briefly describing these different concepts to allow the reader to better understand the big picture of the paper. But the paper is quite well written, and the steps to obtain the final distribution class are clear enough. To help the reader, I would advise the authors to include some parts of the supplementary material directly in the paper, particularly Algorithm 2 and Section B that provide a concrete example of the use of the presented principles. The first paragraph of Section 3 would also benefit from presenting the intuitive ideas behind the obtained distribution class, and particularly the role of the embeddings in the resulting algorithm. Theorem 1 seems to have some typos problems (R^p instead of R^r), and the role of v^* is not clear in this theorem. Section 3.2 is particularly difficult to understand and would benefit from a small description of the MDA principles. The experimental results are interesting and demonstrate the power of the contribution. Of course, a quantitative study on the quality of the generated samples could help to validate the generative ability of the model (using mechanical turk for example). Moreover, the convergence speed is evaluated in term of epochs, while it would be more interesting to evaluate it in term of time since the complexity of each epoch is not clearly stated in the paper, so it is not clear if the proposed approach is really faster than baselines. Summary: interesting contribution, but a paper lacking a clear presentation of the main ideas, making it difficult to read for non-experts, and thus restricted to a small audience.

Reviewer 3



The paper introduces a novel variational algorithm, that aims at dealing highly flexible approximating distributions, in order to reduce to bias of the variational approximation. The paper is obviously interesting but, not being an expert of the field, I found it quite technical and dense, probably due to the lack of space. The central idea of the paper seems to be the reformulation of the minimization of the KL divergence wrt $q$ (the approximate distribution) into an optimization problem wrt a set of 'local variables'. Based on this formulation, the authors derive an optimization algorithm that is reasonably easy to implement. My main comments are about a series of claims that are not obvious to me. * The author describe the iterative transformations as non-parametric and I am not sure to understand in what sense. * The sentence in L163-164 is rather obscure, although it sounds a very general comment on the virtues of the proposed approach. * The proposed optimization embedding relies on the choice of a divergence (denoted $D_\omega$). This choice (and its influence on the results) is not much discussed. Minor points: * L86: should '$q_k$' be '$q_T$'? * L406: 'transform' -> 'transforms'?